# Acute respiratory distress syndrome readmissions: A nationwide cross-sectional analysis of epidemiology and costs of care

**Matthew T. Siuba** [1]*, **Divyajot Sadana**[2], **Shruti Gadre**[1], **David Bruckman**[3], **Abhijit Duggal**[1]

**1** Department of Critical Care Medicine, Cleveland Clinic, Respiratory Institute, Cleveland, Ohio, United States of America, **2** Department of Critical Care Medicine, Sunnybrook Health Sciences Centre, and Interdepartmental Division of Critical Care Medicine, University of Toronto, Toronto, Ontario, Canada, **3** Department of Quantitative Health Sciences, Cleveland Clinic, Center for Populations Health Research, Lerner Research Institute, Cleveland, Ohio, United States of America

* siubam@ccf.org

## Abstract

### Background

Acute Respiratory Distress Syndrome affects approximately 10% of patients admitted to intensive care units internationally, with as many as 40%-52% of patients reporting re-hospitalization within one year.

### Research question/aim

To describe the epidemiology of patients with acute respiratory distress syndrome who require 30-day readmission, and to describe associated costs.

### Study design and methods

A cross-sectional analysis of the 2016 Healthcare Cost and Utilization Project's Nationwide Readmission Database, which is a population-based administrative database which includes discharge data from U.S. hospitals. Inclusion criteria: hospital discharge records for adults age > 17 years old, with a diagnosis of ARDS on index admission, with associated procedure codes for endotracheal intubation and/or invasive mechanical ventilation, who were discharged alive. Primary exposure is adult hospitalization for meeting criteria as described. The primary outcome measure is 30-day readmission rate, as well as patient characteristics and time distribution of readmissions.

### Results

Nationally, 25,170 admissions meeting criteria were identified. Index admission mortality rate was 37.5% (95% confidence interval [CI], 36.2–38.8). 15,730 records of those surviving hospitalization had complete discharge information. 30-day readmission rate was 18.4%, with 14% of total readmissions occurring within 2 calendar days of discharge; these early readmissions had higher mortality risk (odds ratio 1.82, 95% CI 1.05–6.56) compared with

**Data Availability Statement:** All data is available in a public repository, for purchase. https://www.hcup-us.ahrq.gov/databases.jsp. This is third party

data, and available to anyone in the same manner as it was made available to our team.

**Funding:** The author(s) received no specific funding for this work.

**Competing interests:** The authors have declared that no competing interests exist.

**Abbreviations:** APR-DRGs, All Patients Refined Diagnosis Related Group; ARDS, acute respiratory distress syndrome; ICD-10, The International Statistical Classification of Diseases, 10th edition; ICU, intensive care unit; IMV, invasive mechanical ventilation; NRD, Nationwide Readmission Database.

readmission in subsequent days. For the closest all-cause readmission within 30 days, the mean cost was $26,971, with a total national cost of over $75.6 million.

## Interpretation

Thirty-day readmission occurred in 18.4% of patients with acute respiratory distress syndrome in this sample, and early readmission is strongly associated with increased mortality compared to late readmission. Further research is needed to clarify whether the rehospitalizations or associated mortalities are preventable.

## Introduction

The Acute Respiratory Distress Syndrome (ARDS) is characterized by acute lung injury, often a result of pneumonia, sepsis, aspiration, pancreatitis, or trauma. It affects approximately ten percent of patients admitted to intensive care units (ICUs) internationally [1], with U.S. incidence as high as 190,600 cases per year [2]. In-hospital mortality rates range from 38% up to 50% in severe cases [1]. The ongoing burden of healthcare utilization for patients with ARDS is high, with 40%-52% of patients requiring re-hospitalization in one year [3, 4]. In another cohort roughly half of patients required inpatient or post-acute care for 48 days or more after ICU discharge [5].

Previous studies which have explored the epidemiology and risk factors associated with patients with ARDS requiring readmission within 30 days were performed prior to the existence of specific billing codes for ARDS [6, 7]. The most closely associated conditions with ARDS, pneumonia and sepsis, have been previously assessed in nationwide databases, with 30-day readmission rates of 7.5% and 17.5% respectively [8, 9]. Given that a diagnosis of ARDS usually suggests a higher severity of illness, requiring stay in an intensive care unit, we hypothesized that the readmission rates would be higher than these related diseases.

The International Statistical Classification of Diseases, 10th edition (ICD-10), is the first iteration to include a diagnosis code for ARDS [10] and 2016 was the first full calendar year when ICD-10 was implemented. This investigation aimed to describe ARDS readmissions in a large administrative database. Our primary objectives were to define the all-cause 30-day readmission rate for patients with ARDS. Additionally, we describe patient characteristics of those rehospitalized, as well as time distribution of readmissions. Finally, we report financial implications of these readmissions and provide predictors of readmission costs by patient characteristics.

## Methods

### Data source, setting, and participants

The data source for this investigation is the 2016 Healthcare Cost and Utilization Project's (HCUP) Nationwide Readmission Database (NRD), which is drawn from the State Inpatient Databases. The NRD is a population-based administrative database which includes discharge data from U.S. hospitals, accounting for approximately 36 million weighted discharges per year. Twenty-seven states contributed to the database in 2016, accounting for 56.6% of all U.S. hospitalizations. The year 2016 was chosen as it was the first full year where International Statistical Classification of Diseases, 10th edition (ICD-10) was used for billing, which allowed for capture of ARDS admissions [11]. ARDS did not exist as a standalone diagnosis prior to ICD-

10. This study was labeled as exempt by the institutional review board given the NRD is publicly available from HCUP and deidentified. Authors MS and DB completed the HCUP Data Use Agreement Training Course as required.

A cross-sectional analysis of the 2016 NRD was performed. The study population consisted of any hospital admission for an adult age > 17 years old, with a diagnosis of ARDS (ICD-10-CM code: J80) with any associated ICD-10-PCS procedure code(s) for endotracheal intubation and/or invasive mechanical ventilation (IMV) (Codes: 0BH17EZ, 0BH13EZ, 0BH18EZ, 5A1935Z, 5A1945Z, 5A1955Z). We selected for IMV to increase the likelihood of correctly identifying ARDS patients per Berlin criteria (requiring at least 5 centimeters of water of positive end expiratory pressure) [12], as the NRD does not include physiologic measurements or ventilator-specific variables. Index admissions were defined as those admissions discharged alive between January and November 2016 to allow for 30 days of readmission after discharge. Records of patients discharged to a skilled nursing facility or long-term acute care hospitals were included. Charges were converted to costs using appropriate HCUP conversion tables for 2016 [11]. For readmission analyses, complete length of stay information was required. A readmission can be considered as a new index admission when the readmission includes ARDS under the case definition. Therefore, it is possible that a person may have more than one index admission with ARDS and 30-day all-cause readmission. The study cohort was extended to include records meeting the definition of an ARDS index admission or all-cause readmission but where in-hospital mortality occurred.

## Bias

The primary source of bias in this type of investigation includes coding accuracy. Clinicians must not only recognize ARDS (which happens inconsistently [1]), but also document and/or bill for it; thus it is likely that ARDS admissions will be underrepresented. We attempted to account for any overdiagnosis by limiting search to patients who received mechanical ventilation as previously mentioned. Other potential biases, such as discharge of patients to hospitals outside of the NRD database, is possible, but felt unlikely, as it is a reasonably representative sample of US hospitals. Finally, as readmission is likely to be confounded by factors other than admission diagnosis (such as age, biological sex, index admission length of stay, insurance status, and comorbidities, a multivariable), a multivariable logistic regression analysis was performed to account for these factors *a priori*.

## Variables

Patient demographics, comorbidities, discharge disposition and hospital variables associated factors with a 30-day readmission were determined using bivariate testing. Comorbidities selected *a priori* were included in multivariable logistic models adjusted for the survey weights. Furthermore, multivariable linear regression was performed to model factors associated with (1) index admission length of stay, (2) cost of index admission, and (3) cost of readmission. Analyses were performed using SAS PROC SURVEY procedures to adjust for the complex survey design weights (SAS Institute, Inc., Cary, NC, USA). Study and manuscript preparation followed the recommendations of the EQUATOR network's STROBE (The Strengthening the Reporting of Observational Studies in Epidemiology) guidelines for observational studies [13].

## Outcomes

The primary outcome was 30-day readmission rate for all-cause readmissions. Total index admissions, total readmissions, time to first readmission, and costs associated with admission and readmission (among states and sites providing adequate charge and cost data) were also

captured. A key secondary outcome was early readmissions occurring within 2 days of discharge. Incidence of ARDS cases for all 12 months of 2016 was also assessed, as well as the index and readmission mortality rates. To study etiologies for admission as well as other measures of association, the listing of All Patients Refined Diagnosis Related Groups (APR-DRGs) and comorbidities as identified using methods recommended for administrative data sets using ICD coding [14–16]. The comorbidities were defined using SAS coding provided by HCUP based on Elixhauser definitions, but specific comorbidities were selected *a priori* for multivariable logistic regression modeling. Demographic data such as age groups, biological sex, expected payor, and discharge disposition were collected. Mean and median costs for index admission as well as readmission were obtained. Potentially relevant procedures such as tracheostomy and extracorporeal life support were also queried.

## Statistical methods

Because the NRD is based on the complex survey design of the HCUP data, weights reflecting the sampling distribution of strata and clusters are required. Admissions are weighted up to the total admissions occurring in the non-institutionalized US population in 2016; as such, all presented counts are weighted values unless noted. Additional details can be found at the HCUP website [17]. Summary statistics including means and standard deviations for continuous variables as well as counts and percentages for categorical variables were used to describe the study cohort. Bivariate analyses were performed to determine the association between outcomes and potential explanatory variables. Categorical variables were assessed with Rao-Scott chi-square test, and continuous variables were compared using analysis of variance tests adjusted for the sampling design. Several APR-DRGs were assessed for association with readmission risk, and Tukey corrections were applied to account for pairwise contrasts. Bonferroni corrections for multiple comparisons were not applied due to efficiency of the sampling design and weighting; application of this correction could overcompensate for the overall significance (alpha) level [18, 19].

## Results

### Patient characteristics

Overall, more than 35.6 million total admission records were identified in 2016. Of those, 83,212 admissions with any ICD-10 code for ARDS (J80) were identified, with 25,170 meeting inclusion criteria. Demographic information for these groups is included in the S1 Table. In total, 15,730 (95% confidence interval [CI] 14,837–16,624) admissions contained complete length of stay information, surviving to discharge, and thus were eligible for readmission analysis. Of the 25,170 hospitalizations, 9,439 ended in death, with index admission mortality rate 37.5% (95% CI 36.2–38.8). Index hospitalization mortality stratified by age are included in S2 Table. Demographic information on records meeting case definition, stratified by index hospitalization mortality, are presented in Table 1. No patients in the index admission or readmission samples were identified as having received extracorporeal life support despite a query for those billing codes.

### Outcomes

The national estimate of index admissions with at least one readmission within 30 days is 2,889 (95% CI 2,656–3,122), reflecting 18.4% (95% CI 17.4–19.3) of all eligible index admissions. The median time from index discharge to closest readmission is 10.6 days (95% CI 9.9–11.3). Time distribution of readmissions is shown in Fig 1. Notably, 14.3% (95% CI 11.5–17.1)

**Table 1. Demographics: Index events and mortality.**

| Factor | N | Survived Index Admission (N = 15,730) | In-Hospital mortality (N = 9,439) | p-value |
|---|---|---|---|---|
| Indicator of sex, % (95% CI) | 25,170 | | | 0.15[c] |
| Male | 13,208 | 51.9 (50.6, 53.2) | 53.4 (51.9, 54.9) | |
| Female | 11,962 | 48.1 (46.8, 49.4) | 46.6 (45.1, 48.1) | |
| Primary expected payer (uniform), % (95% CI)* | 25,132 | | | *<0.001*[c] |
| Medicare | 12,501 | 46.4 (44.8, 47.9) | 55.4 (53.3, 57.5) | |
| Medicaid | 4,944 | 21.2 (19.8, 22.6) | 17.2 (15.7, 18.6) | |
| Prvt. Ins/HMO | 5,836 | 25.0 (23.6, 26.3) | 20.3 (18.8, 21.9) | |
| Self-pay | 947.5 | 4.0 (3.4, 4.5) | 3.5 (2.8, 4.1) | |
| No Charge | 106.7 | 0.52 (0.29, 0.75) | 0.26 (0.08, 0.44) | |
| Other | 795.9 | 3.0 (2.6, 3.5) | 3.4 (2.7, 4.1) | |
| Patient Location: NCHS Urban-Rural Code, % (95% CI)* | 25,093 | | | 0.55[c] |
| Large Central Metro | 6,509 | 26.7 (24.2, 29.2) | 24.7 (22.1, 27.2) | |
| Large Fringe Metro | 6,199 | 24.5 (22.0, 27.0) | 25.0 (21.4, 28.6) | |
| Medium Metro | 5,132 | 20.0 (17.8, 22.3) | 21.2 (18.7, 23.6) | |
| Small Metro | 2,464 | 9.9 (8.5, 11.2) | 9.8 (8.3, 11.2) | |
| Micropolitan | 2,607 | 10.3 (8.9, 11.6) | 10.6 (9.4, 11.8) | |
| Noncore | 2,181 | 8.6 (7.5, 9.7) | 8.8 (7.2, 10.4) | |
| Elective versus non-elective admission, % (95% CI)* | 25,124 | | | 0.36[c] |
| No | 23,095 | 91.7 (90.9, 92.5) | 92.3 (91.1, 93.4) | |
| Yes | 2,029 | 8.3 (7.5, 9.1) | 7.7 (6.6, 8.9) | |
| Median household income national quartile for patient ZIP Code, % (95% CI)* | 24,789 | | | *0.022*[c] |
| First quartile | 7,990 | 32.9 (30.8, 34.9) | 31.2 (28.7, 33.7) | |
| Second quartile | 6,442 | 26.6 (24.9, 28.3) | 24.9 (22.9, 26.9) | |
| Third quartile | 5,890 | 23.0 (21.6, 24.4) | 25.1 (23.3, 26.9) | |
| Fourth quartile | 4,465 | 17.5 (15.8, 19.3) | 18.8 (16.7, 21.0) | |

*Data not available for all subjects, as unweighted frequencies: Primary expected payer (uniform) = 20; Patient Location: NCHS Urban-Rural Code = 48; Elective versus non-elective admission = 18; Median household income national quartile for patient ZIP Code = 214.

Frequencies presented are weighted counts. P-values:

[a] = linear regression;

[b] = linear regression with log transformation;

[c] = Rao-Scott chi-square test.

of readmissions occurred within the first two calendar days after discharge; 37.1% (95% CI 28.1–44.4) occurred within 1 week. Early (day 1–2) vs late (day 3–30) readmissions were associated with higher risk of in-hospital mortality (odds ratio [OR] 1.83, 95% CI 1.15–2.91). Readmission mortality rate was 8.45% (95% CI 6.7–10.0); results stratified by age are reported in S2 Table. Most prevalent APR-DRGs are reported in Table 2. Diagnoses related to sepsis and/or infections make up approximately 23% of APR-DRGs; 18.3% of APR-DRGs involved mechanical ventilation for more than 96 hours' duration, with the majority being associated with a tracheostomy in place.

## Risk factors for readmission

A multivariable logistic regression was performed using covariates selected *a priori* to predict risk of readmission, with results in Table 3. Notably, length of stay, increasing age, payer status,

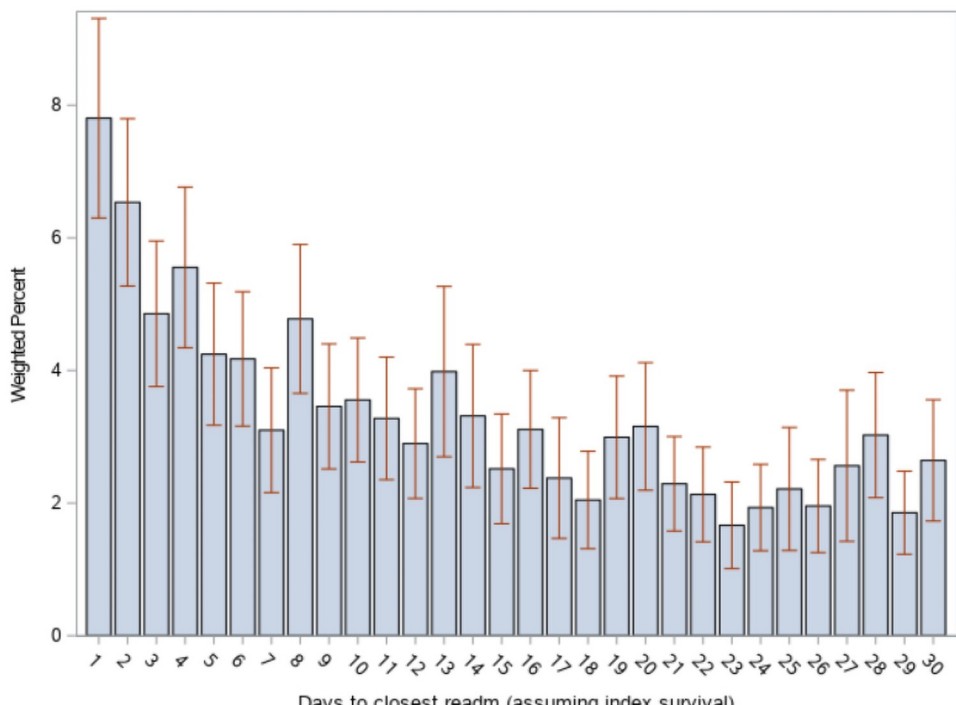

**Fig 1. Distribution of first readmission over 30 days.** X-axis: Day after discharge from index admission. Bars indicate 95% confidence intervals.

and several comorbidities were associated with increased readmission risk. Urban-rural patient locations other than large fringe metro or medium metro were associated with decreased risk of readmission. The model had very modest predictive value (C statistic 0.599) but did perform better than a model lacking covariates (Likelihood ratio test p < 0.0001). Adding discharge disposition to this model did not improve its performance. However, in a bivariate association of index discharge disposition and readmission, there was a statistically detectable difference between those readmitted and not readmitted, with the largest differences

**Table 2. Top 10 all patient related diagnostic related groups at index admission and readmission.**

| Coding description | Index Admission | | Readmission | |
|---|---|---|---|---|
| | N (SD) | % (SE) | N (SD) | % (SE) |
| Septicemia & disseminated infections | 2787 (118.5) | 17.7 (0.57) | 437 (33.6) | 15.2(1.00) |
| Infectious & parasitic diseases including HIV | 1055 (66.0) | 6.7 (0.35) | 225 (26.2) | 7.8 (0.81) |
| Heart failure | 728 (52.3) | 4.6 (0.31) | 193 (21.6) | 6.7 (0.70) |
| Other respiratory diagnoses except signs, symptoms & minor diagnoses | 543 (38.3) | 3.4 (0.22) | 114 (14.7) | 3.9 (0.4) |
| Chronic obstructive pulmonary disease | 534 (45.5) | 3.4 (0.27) | 134 (17.5) | 4.6 (0.57) |
| Respiratory failure | 430 (36.9) | 2.7 (0.22) | 100 (19.1) | 3.5 (0.63) |
| Other pneumonia | 417 (29.7) | 2.6 (0.19) | 80 (11.9) | 2.8 (0.42) |
| Poisoning of medicinal agents | 230 (22.9) | 1.5 (0.14) | 36 (8.9) | 1.2 (0.30) |
| Asthma | 214 (39.5) | 1.4 (0.25) | 33 (9.7) | 1.1 (0.33) |
| Other disorders of nervous system | 197 (26.1) | 1.3 (0.17) | --- | --- |

Ordered by descending frequency on readmission. Weighted values unless otherwise specified. SD = standard deviation, SE = standard error.

**Table 3. Risks of 30-day readmission.**

| Factor | Odds Ratio Estimate | 95% Confidence Limits |
|---|---|---|
| LOS (effect of one additional day) | **1.005** | **(1.002, 1.007)** |
| Male (vs Female) | 0.902 | (0.804, 1.013) |
| Age Group (ref: 75 and over) | | |
| 18-34y | 0.982 | (0.744, 1.296) |
| 35-44y | 0.988 | (0.751, 1.299) |
| 45-54y | 1.18 | (0.934, 1.491) |
| 55-64y | **1.284** | **(1.044, 1.580)** |
| 65-74y | **1.218** | **(1.011, 1.468)** |
| Primary Payer (ref: Private Ins) | | |
| Medicare | **1.833** | **(1.202, 2.797)** |
| Medicaid | **1.617** | **(1.076, 2.432)** |
| Self-Pay | 1.504 | (0.544, 4.158) |
| No Charge | 1.618 | (0.920, 2.844) |
| Other | **1.558** | **(1.073, 2.263)** |
| NCHS Urban-Rural Patient Location (ref: Large Central Metro) | | |
| Large Fringe Metro | 0.845 | (0.709, 1.006) |
| Medium Metro | 0.846 | (0.667, 1.073) |
| Small Metro | **0.768** | **(0.603, 0.979)** |
| Micropolitan | **0.716** | **(0.544, 0.940)** |
| Noncore | **0.859** | **(0.758, 0.973)** |
| Comorbidities | | |
| Peripheral vascular disease | **1.284** | **(1.062, 1.552)** |
| CHF | 1.137 | (0.985, 1.313) |
| Chronic pulmonary disease | **1.194** | **(1.049, 1.359)** |
| DM w/o chronic complic. | 0.993 | (0.837, 1.178) |
| DM with chronic complications | 1.068 | (0.912, 1.250) |
| Renal failure | **1.299** | **(1.106, 1.526)** |
| Liver disease | 1.158 | (0.958, 1.400) |
| Any Cancer history | **1.296** | **(1.023, 1.641)** |
| Hypertension | 1.009 | (0.869, 1.172) |
| Deficiency Anemias | **1.211** | **(1.064, 1.377)** |

Adjusted odds ratio and 95% confidence intervals are presented.

being a higher proportion of discharge to "other facility" and lower proportion of discharge to home or unknown in those readmitted (S3 Table).

Bivariate analysis of seven APR-DRGs selected *a priori*, adjusted for multiple comparisons, demonstrated a statistically detectable increase in the proportion of patients with sepsis (but no other condition) on index admission in those readmitted compared to those who were not (S4 Table). Of eligible readmissions, 16.1% (95% CI 15.0–17.2) included a procedure code for tracheostomy. Receiving a tracheostomy on index admission did not detectably increase risk of readmission, occurring in 16.0% (95% CI 14.8–17.1) of those readmitted vs 16.4% (95% CI 14.3–18.5) who were not, p = 0.69.

## Cost analysis

The national estimate for mean cost of an ARDS index admission was $71,004 in 2016 dollars, not adjusted for inflation, excluding deaths, with a total national cost of over $1.09 billion.

**Table 4. Total costs among index admissions (deaths excluded) and the closest readmission—Weighted statistics.**

| Parameter | Mean* | 95% C.I. | Sum | 95% C.I. |
|---|---|---|---|---|
| Total Costs | $71,004 | (67,425–74,583) | $1,089,415,110 | (997,796,632–1,181,033,588) |
| Total readmission cost for the closest readmission* | $26,971 | (24,186–29,756) | $75,554,244 | (66,523,019–84,585,470) |
| For Age 18 – 64Y: | | | | |
| Total Cost | $81,152 | (76,658–85,646) | $803,406,632 | (724,996,733–881,816,531) |
| Total Readmissions cost for the closest readmission | $28,674 | (24,733–32,615) | $49,411,171 | (41,604,882–57,217,459) |
| For Age 65Y and older: | | | | |
| Total Cost | $52,547 | (49,475–55,619) | $286,008,478 | (262538696–309478260) |
| Total Readmissions cost for the closest readmission | $24,250 | (20,872–27,628) | $26,143,073 | (21,783,113–30,503,034) |

Details of cost analysis are presented in Table 4. For the closest all-cause readmission within 30 days, the mean cost was $26,971, with a total national cost of over $75.6 million. When stratified by age (less than 65 years old versus 65 and above), mean index admission cost was substantially higher in the younger group than the older group (point estimate for the difference $28,605), but not detectably different on readmission costs. Analysis of the difference in index admission costs by age demonstrated that differences in cost by age were driven by the inverse correlation of age with length of stay (S5 Table).

Analysis of readmission costs demonstrated substantially higher costs for early (days 1–2) readmission as well as for those who died on readmission. A regression model for total cost on readmission was developed that included length of stay, sex, insurance status, urban/rural location, age group, early admission, mortality, and comorbidities. Notably, the model estimated additional cost for early readmission was $16,919 (S6 Table). The wide gap between costs based on mortality, stratified by age, are presented in S1 Fig. More details of this regression analysis are included in the supplement section titled "Supplementary Analysis: Regression Model for Readmission Costs." An example charge calculation using this model is included in S7 Table.

## Discussion

In the largest study of its kind, we demonstrate that 30-day readmission occurs in nearly one fifth of patients admitted with acute respiratory distress syndrome. More than 14% of these readmissions occur in the first two calendar days after discharge, and these early readmissions are associated with 83% higher risk of mortality compared to readmissions in the subsequent 28 days. The distribution of most commonly associated diagnoses did not meaningfully change from index admission to readmission, as shown in Table 2. Risk of readmission was increased with length of stay, increasing age, non-private insurance (Medicare and Medicaid), and comorbidities such as a history of renal dysfunction, malignancy, anemia, chronic pulmonary disease, and anemia. The mean cost for ARDS index admission and readmission were over $71,004 and $26,971, respectively, for annual total costs over $1.09 billion and $75.6 million, respectively. Costs on index admission and readmission were also higher with decreasing age and increasing length of stay. Notably, readmission within 2 calendar days of discharge, as well as dying during readmission were associated with substantially higher costs.

Prior studies which utilized codes for acute hypoxemic respiratory failure to evaluate ARDS readmission risk found 30-day rates between 12–18% [6, 7], similar to our findings. The readmission rate is higher than previously reported for pneumonia [8], but nearly identical to rates seen in heart failure and sepsis readmission studies [9, 20]. Total charges were substantially higher than seen in a previous study of sepsis ($4.2 vs 3.5 billion), with a similar median time

to readmission [9]. It is surprising that readmission rates and costs were not even higher in our analysis, as we selected a sicker cohort group of patients by including only those who required invasive mechanical ventilation. Finally, while we were not able to examine prior healthcare utilization specifically, our multivariable model supports a higher risk of readmission for those with certain comorbidities and increasing age, similar to prior work in more granular datasets [21].

This analysis revealed important information on readmissions within the first two days after discharge, which were associated with significantly higher readmission mortality risk as well as costs. Future research should address whether these early readmissions represent an opportunity to improve discharge risk stratification or planning. The distribution of most frequently billed codes did not appear to meaningfully change from index admission to readmission, which suggests the possibility the original condition recurred or did not sufficiently remit in the first place. Additionally, with an average cost of readmission of almost $27,000, readmission prevention measures should be investigated and prospectively tested. Costs to the healthcare system are significant, and ARDS patients have been shown to be particularly susceptible to financial toxicity related to medical bills, insurance loss, and change in employment status [22].

We observed that younger patients incurred a higher cost on readmission, markedly so in those who died during readmission. This may be due to a higher propensity to escalate and sustain (rather than withdraw) aggressive care measures in younger people compared to those who were older. Of note, we did not capture any records of patients in this study who received extracorporeal life support. We did detect a longer length of stay in younger patients as well which could be the driver of the increased costs, for the same rationale of increased life-sustaining care. Further breakdown in the specific charges (in terms of services rendered) which led to the total costs was not possible in this database, however, beyond the data we presented.

It is unclear what effect, if any, readmissions within the first month after discharge have on the overall disease recovery trajectory in ARDS. Long-term sequelae such as physical and neurocognitive dysfunction which can persist from months to years are well described in the literature [23–25]. The nature of the administrative database used in our analysis does not allow for assessment of physical or neurocognitive function; further research could explore whether readmissions signify increased risk for these conditions which could potentially benefit from more intensive rehabilitation and other risk modification.

## Limitations

Any investigation based on administrative datasets is subject to meaningful limitations. At minimum, coding data has issues with accuracy [26]. ARDS itself is prone to misclassification by clinicians, with it being unrecognized up to 40% of the time [1]. Perhaps most importantly, the NRD lacks granular clinical information such as lab data, imaging, and details on the provision of lung-protective mechanical ventilation, non-invasive respiratory supports, other organ support modalities. It is worth noting that no diagnostic codes identified mentioned circulatory shock which is reasonably prevalent in patients with ARDS. Finally, no patients who received extracorporeal life support were identified, which likely leads to an underestimation of overall costs, mortality, and readmission rate. Our query only showed 339 weighted admissions containing codes for extracorporeal support in the entirety of the 2016 NRD.

The 2016 NRD was the first time that a full year of ICD-10 codes, and therefore specific coding for ARDS, was available, which makes putting findings in context more challenging. However, the index mortality rate of roughly 38% corresponds to other large observational studies of ARDS [1, 27, 28]. Additionally, the associated diagnostic codes associated with

readmission are indeed commonly seen in ARDS, especially sepsis or other infectious etiologies [1]. The ability to query a large, multi-state database provided an opportunity to evaluate general patterns across the United States.

## Conclusion

Thirty-day readmission occurred in 18.4% of patients with acute respiratory distress syndrome in this sample, and early readmission is strongly associated with increased mortality and cost compared to late readmission. Further research is needed to clarify whether the rehospitalizations or associated mortalities are preventable.

## Supporting information

**S1 Table. Demographics on all NRD records vs case definitions.**
(DOCX)

**S2 Table. Age-stratified mortality of records meeting case definition during index admission and readmission.**
(DOCX)

**S3 Table. Disposition of index admission: Bivariate association of disposition and readmission.**
(DOCX)

**S4 Table. Bivariate association of APR-DRG of the index admission and 30-day readmission.**
(DOCX)

**S5 Table. Index admission total cost and length of stay (LOS) differences across age groups for those with a readmission.** All results are weighted. Total costs are unadjusted.
(DOCX)

**S6 Table. Readmission total cost regression modeling estimates.**
(DOCX)

**S7 Table. Example charge calculation.**
(DOCX)

**S1 Fig. Age and mortality coefficient for readmission cost regression model.**
(DOCX)

## Acknowledgments

The authors thank the Healthcare Cost and Utilization Project Data Partners that contributed www.hcup-us.ahrq.gov/hcupdatapartners.jsp. The authors would also like to thank the Center for Populations Health Research (CPHR) and the Lerner Research Institute leadership at the Cleveland Clinic for providing analytical support through the Collaboration Research Award.

## Author Contributions

**Conceptualization:** Matthew T. Siuba, Divyajot Sadana, Shruti Gadre, David Bruckman, Abhijit Duggal.

**Data curation:** David Bruckman.

**Formal analysis:** Matthew T. Siuba, Shruti Gadre, David Bruckman, Abhijit Duggal.

**Investigation:** Matthew T. Siuba, Divyajot Sadana, David Bruckman, Abhijit Duggal.

**Methodology:** Matthew T. Siuba, David Bruckman, Abhijit Duggal.

**Supervision:** Abhijit Duggal.

**Writing – original draft:** Matthew T. Siuba, David Bruckman.

**Writing – review & editing:** Matthew T. Siuba, Divyajot Sadana, David Bruckman, Abhijit Duggal.

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
