## [Decision Letter · Decision Letter 0]

27 Sep 2021

PONE-D-21-13742Acute respiratory distress syndrome readmissions: a nationwide cross-sectional analysis of epidemiology and costs of carePLOS ONE

Dear Dr. Siuba,

Thank you for submitting your manuscript to PLOS ONE. After careful consideration, we feel that it has merit but does not fully meet PLOS ONE’s publication criteria as it currently stands. Therefore, we invite you to submit a revised version of the manuscript that addresses the points raised during the review process.

We look forward to receiving your revised manuscript.

Kind regards,

Brenda M. Morrow, PhD

Academic Editor

PLOS ONE

Journal Requirements:

4. We note you have included a table to which you do not refer in the text of your manuscript. Please ensure that you refer to Table 7e in your text; if accepted, production will need this reference to link the reader to the Table.

Reviewers' comments:

Reviewer's Responses to Questions

**Comments to the Author**

1. Is the manuscript technically sound, and do the data support the conclusions?

Reviewer #1: Yes

Reviewer #2: Yes

2. Has the statistical analysis been performed appropriately and rigorously? 

Reviewer #1: Yes

Reviewer #2: Yes

3. Have the authors made all data underlying the findings in their manuscript fully available?

Reviewer #1: Yes

Reviewer #2: Yes

4. Is the manuscript presented in an intelligible fashion and written in standard English?

Reviewer #1: Yes

Reviewer #2: Yes

5. Review Comments to the Author

Reviewer #1: This manuscript presents the results of a cross-sectional analysis of a large, national, administrative database to evaluate ARDS. This study is the first of its kind to use data after the inclusion of ARDS as a distinct ICD-10 code, potentially improving its detection in administrative datasets.

Strengths of the study include:

1) The findings of the study have strong face validity, with admission information, associated diagnoses, and mortality rate all in line with previous prospectively collected data. It is highly likely that the dataset has accurately identified the population of interest.

2) Statistical methods are clearly explained and are rigorous and appropriate for the use of the administrative dataset. Choices are explained thoroughly.

3) The writing is clear and concise, and the tables and figures add to the manuscript.

Limitations of the study include:

1) The findings (or lack thereof) of ECLS in the database are concerning, though perhaps not surprising. Survival for ARDS requiring ECLS is lower than for those without. The authors only identified 339 cases of ECLS in the entire database for the year, suggesting a significant under-reporting. However, this does limit the potential generalizability to ARDS findings in general. Future work could include a possible link with ELSO database or analysis of later years within the NRD to identify trends in ECLS coding.

2) Page 5, Line 168: This sentence initially led me to think the patient had to also survive the readmission, however after further reading in the manuscript, it became clear this is not the case. I would advise simply clarifying this sentence to avoid confusion of the reader.

Reviewer #2: Please see attached file.

In this paper, Dr Siuba and colleagues have used the HCUP’s NRD, which contains discharge abstract data for over 36 million weighted discharges, to describe the rate of 30-day readmission after an admission to hospital with a diagnosis of ARDS (defined using ICD-10 codes and procedure codes for intubation or tracheostomy). They also sought to explore the costs associated with both the admission to hospital for ARDS and a subsequent readmission. They identified 25,170 index episodes, with 9,439 dying during index admission, and 2889 experiencing a readmission within 30 days. 14.3% of the readmissions occurred within the first two days following discharge, and most had codes for the same most responsible diagnosis as was noted for the index admission. They identified some predictors of readmission using a multivariable logistic regression model that notably demonstrated that older age, and longer stay in hospital were associated with higher hospital costs. They found that the mean cost of hospitalization for ARDS was $71,004, and the mean cost of readmission was 26,971. Costs stratified by age demonstrated that younger groups had higher costs as well as longer mean length of stays than older groups. They also conducted a regression analysis to predict readmission costs based on various covariates.

6. PLOS authors have the option to publish the peer review history of their article (what does this mean?). If published, this will include your full peer review and any attached files.

Reviewer #1: **Yes: **Thomas Bice, MD, MSc

Reviewer #2: No

---

## [Author Response · Author response to Decision Letter 0]

23 Nov 2021

Dear Drs. Morrow, Bice, and Reviewer #2,

Thank you kindly for the opportunity to revise this submission. Within the limitations of what can be achieved with this data set and timing of revision, we attempted to address every possible concern raised during the first review. We feel this version is improved significantly with your input and look forward to your feedback.

Matthew Siuba, DO

On behalf of the authors

Reponse to Editor Comments:

1. All sections were formatted according to PLOS ONE guidelines.

With regards to the sharing of the de-identified data set, this information has been clarified in the main manuscript. HCUP NRD database is available on request from the publisher (HCUP), not from our team directly. The usage of the dataset requires completion of a data use agreement and fee paid to HCUP.

b) If there are no restrictions, please upload the minimal anonymized data set necessary to replicate your study findings as either Supporting Information files or to a stable, public repository and provide us with the relevant URLs, DOIs, or accession numbers. For a list of acceptable repositories, please see http://journals.plos.org/plosone/s/data-availability#loc-recommended-repositories. We will update your Data Availability statement on your behalf to reflect the information you provide.

Please see the answer in 2a above.

Thank you, captions have been added to the end of the manuscript, before references.

4. We note you have included a table to which you do not refer in the text of your manuscript. Please ensure that you refer to Table 7e in your text; if accepted, production will need this reference to link the reader to the Table.

A reference to Table 7e (now S7 Table) is included at the end of the results section.

 

Reviewer #1: This manuscript presents the results of a cross-sectional analysis of a large, national, administrative database to evaluate ARDS. This study is the first of its kind to use data after the inclusion of ARDS as a distinct ICD-10 code, potentially improving its detection in administrative datasets.

Strengths of the study include:

1) The findings of the study have strong face validity, with admission information, associated diagnoses, and mortality rate all in line with previous prospectively collected data. It is highly likely that the dataset has accurately identified the population of interest.

2) Statistical methods are clearly explained and are rigorous and appropriate for the use of the administrative dataset. Choices are explained thoroughly.

3) The writing is clear and concise, and the tables and figures add to the manuscript.

Thank you very much for the kind comments, we appreciate your attention to our choices and rigor.

Limitations of the study include:

1) The findings (or lack thereof) of ECLS in the database are concerning, though perhaps not surprising. Survival for ARDS requiring ECLS is lower than for those without. The authors only identified 339 cases of ECLS in the entire database for the year, suggesting a significant under-reporting. However, this does limit the potential generalizability to ARDS findings in general. Future work could include a possible link with ELSO database or analysis of later years within the NRD to identify trends in ECLS coding.

We agree that this is a significant limitation and the directions for future study mentioned are intriguing. In our discussion, we state:

“Finally, no patients who received extracorporeal life support were identified, which likely leads to an underestimation of overall costs, mortality, and readmission rate.”

2) Page 5, Line 168: This sentence initially led me to think the patient had to also survive the readmission, however after further reading in the manuscript, it became clear this is not the case. I would advise simply clarifying this sentence to avoid confusion of the reader.

Thank you for the important clarification. The sentence was revised:

“For readmission analyses, complete length of stay information was required”

 

Reviewer #2 Comments

In this paper, Dr Siuba and colleagues have used the HCUP’s NRD, which contains discharge abstract data for over 36 million weighted discharges, to describe the rate of 30-day readmission after an admission to hospital with a diagnosis of ARDS (defined using ICD-10 codes and procedure codes for intubation or tracheostomy). They also sought to explore the costs associated with both the admission to hospital for ARDS and a subsequent readmission. They identified 25,170 index episodes, with 9,439 dying during index admission, and 2889 experiencing a readmission within 30 days. 14.3% of the readmissions occurred within the first two days following discharge, and most had codes for the same most responsible diagnosis as was noted for the index admission. They identified some predictors of readmission using a multivariable logistic regression model that notably demonstrated that older age, and longer stay in hospital were associated with higher hospital costs. They found that the mean cost of hospitalization for ARDS was $71,004, and the mean cost of readmission was 26,971. Costs stratified by age demonstrated that younger groups had higher costs as well as longer mean length of stays than older groups. They also conducted a regression analysis to predict readmission costs based on various covariates. 

The study question is important, and well-justified. 

Thank you for the summary and commentary.

Major comments: 

1. I would suggest that the authors rearrange the methods section slightly so that it follows more closely the EQUATOR checklist. I don’t see any descriptions of considerations of bias, or confounding, and these should be addressed. 

Thank you very much for pointing this out. The methods section has been rewritten accordingly to follow the EQUATOR STROBE checklist order as much as possible, without compromising clarity.

2. The authors have identified some interesting trends, particularly that costs were higher among younger patients, who also had longer lengths of stay. Do the authors have any thoughts on how age and length of stay might interact in the causal pathway to costs? 

We agree that this is an interesting finding. Please see the following commentary added in the discussion section:

“We observed that younger patients incurred a higher cost on readmission, markedly so in those who died during readmission. This may be due to a higher propensity to escalate and sustain (rather than withdraw) aggressive care measures in younger people compared to those who were older. Of note, we did not capture any records of patients in this study who received extracorporeal life support. We did detect a longer length of stay in younger patients as well which could be the driver of the increased costs, for the same rationale of increased life-sustaining care. Further breakdown in the specific charges (in terms of services rendered) which led to the total costs was not possible in this database, however, beyond the data we presented.”

3. Does the database only provide the total cost for an admission, or are the costs broken down into categories of costs? If they can be broken down, I would suggest describing the breakdown for the total cohort, as well as for the group that die during index admission, discharged with no readmission and those with a readmission. You might want to also describe costs for readmission stratified by early versus later readmission. 

This is a great point. Unfortunately, we only see the total reported charges in this type of administrative dataset, so further cost breakdown is not possible. This limitation has been added to the discussion section as mentioned in 2. Above.

4. A significant limitation of this study is that health care utilization costs and readmission are likely predicted by previous healthcare use. Previous research has demonstrated that high-users of healthcare, and those who have higher health care costs (High cost users) are likely to remain high-cost users. Can they do a look back and identify previous healthcare utilization and costs for their cohort, and then either stratifiy or control for this in their models? Previous work (eg: https://www.cmaj.ca/content/188/3/182) has demonstrated that of 45% of high-cost users remain high-cost users over the next two years. 

Thank you for the insightful comment. This would be helpful but is not possible with the limitations of this particular dataset. We did demonstrate in the multivariable model (Table 3), however, that increased age and certain comorbidities made readmission more likely, which supports a similar message.

A comment including your suggested citation is included in paragraph 2 of the discussion:

“Finally, while we were not able to examine prior healthcare utilization specifically, our multivariable model supports a higher risk of readmission for those with certain comorbidities and increasing age, similar to prior work in more granular datasets[21].”

5. The authors have described total costs, but have not described the attributable costs of ARDS. These costs are heavily confounded by the costs associated with underlying medical comorbidities, previous healthcare spending, hospitalization in general, and are not described relative to the costs associated for other non-ARDS types of hospitalizations. They might want to consider a matched analysis comparing costs of hospitalization for ARDS to a group hospitalized for other causes. (See this as an example of costing study employing this methodology: 10.1097/CCM.0000000000004777)

Thank you for the comment. We agree this would be a valuable comparison to make. Our linear regression cost model (see the supplement and specifically S6 Table) supports that comorbidities, insurance status, and age all impact the cost of readmissions. Unfortunately the statistical funding support for this study has ended so further analysis (e.g. with propensity matching) as suggested is not feasible at this time.

Minor: 

1. Is it possible that patients were readmitted to a hospital that might not have reported discharge abstract data to this data set, and thus had a missed readmission? Or does the dataset include all data from all hospitals? 

This is absolutely a concern, given the sampling procedure of HCUP’s NIS and NRD datasets. Fortunately, hospitals which are not eligible for HCUP reporting comprise a very small percentage of US hospitals. A note above this has been added to the new “Bias” section in the Methods section.

2. There is no mention of the cost findings in either the take-home section or the abstract. 

Thank you for pointing this out. The take-home section was removed to meet the formatting of the journal. The results section of the abstract was updated to mention readmission costs.

---

## [Decision Letter · Decision Letter 1]

11 Jan 2022

Acute respiratory distress syndrome readmissions: a nationwide cross-sectional analysis of epidemiology and costs of care

PONE-D-21-13742R1

Dear Dr. Siuba,

We’re pleased to inform you that your manuscript has been judged scientifically suitable for publication and will be formally accepted for publication once it meets all outstanding technical requirements.

Kind regards,

Brenda M. Morrow, PhD

Academic Editor

PLOS ONE

Additional Editor Comments: The second reviewer was not available to re-review, but as the handling editor, I have reviewed the author responses and, together with reviewer #1 am satisfied with the responses and changes made.

Reviewer's Responses to Questions

**Comments to the Author**

1. If the authors have adequately addressed your comments raised in a previous round of review and you feel that this manuscript is now acceptable for publication, you may indicate that here to bypass the “Comments to the Author” section, enter your conflict of interest statement in the “Confidential to Editor” section, and submit your "Accept" recommendation.

Reviewer #1: All comments have been addressed

2. Is the manuscript technically sound, and do the data support the conclusions?

Reviewer #1: Yes

3. Has the statistical analysis been performed appropriately and rigorously? 

Reviewer #1: Yes

4. Have the authors made all data underlying the findings in their manuscript fully available?

Reviewer #1: Yes

5. Is the manuscript presented in an intelligible fashion and written in standard English?

Reviewer #1: Yes

6. Review Comments to the Author

Reviewer #1: Thank you for addressing my comments and the comments of the other reviewers. Thank you for your important work.

7. PLOS authors have the option to publish the peer review history of their article (what does this mean?). If published, this will include your full peer review and any attached files.

Reviewer #1: **Yes: **Thomas Bice, MD, MSc

---

## [Editor Report · Acceptance letter]

13 Jan 2022

PONE-D-21-13742R1 

Acute respiratory distress syndrome readmissions: a nationwide cross-sectional analysis of epidemiology and costs of care 

Dear Dr. Siuba:

I'm pleased to inform you that your manuscript has been deemed suitable for publication in PLOS ONE. Congratulations! Your manuscript is now with our production department. 

Kind regards, 

on behalf of

Professor Brenda M. Morrow 

Academic Editor

PLOS ONE